# Polyphenolics with Strong Antioxidant Activity from *Acacia nilotica* Ameliorate Some Biochemical Signs of Arsenic-Induced Neurotoxicity and Oxidative Stress in Mice

**DOI:** 10.3390/molecules27031037

**Published:** 2022-02-03

**Authors:** Tahira Foyzun, Abdullah Al Mahmud, Md. Salim Ahammed, Md. Imran Nur Manik, Md. Kamrul Hasan, KM Monirul Islam, Simin Sobnom Lopa, Md. Yusuf Al-Amin, Kushal Biswas, Mst. Rejina Afrin, AHM Khurshid Alam, Golam Sadik

**Affiliations:** 1Department of Pharmacy, Southeast University, Dhaka 1212, Bangladesh; tahirafoyzun87@gmail.com; 2Department of Pharmacy, University of Rajshahi, Rajshahi 6205, Bangladesh; littleru21@gmail.com (A.A.M.); salimahamad2017@gmail.com (M.S.A.); moniferdous@yahoo.com (K.M.I.); siminshabnam@yahoo.com (S.S.L.); yusufrupharma@yahoo.com (M.Y.A.-A.); khurshid.jaist@gmail.com (A.K.A.); 3Department of Pharmacy, Northern University Bangladesh, Dhaka 1205, Bangladesh; imran.md39@gmail.com; 4Department of Pharmacy, Comilla University, Kotbari, Cumilla 3506, Bangladesh; kh.shikhon22@gmail.com; 5Department of Pharmacy, East West University, Dhaka 1212, Bangladesh; kushal71@outlook.com (K.B.); rejinaafrin@ymail.com (M.R.A.)

**Keywords:** arsenic, neurotoxicity, biomarker, *Acacia nilotica*, phenolics, flavonoids, acetylcholinesterase, antioxidant activity

## Abstract

Neurotoxicity is a serious health problem of patients chronically exposed to arsenic. There is no specific treatment of this problem. Oxidative stress has been implicated in the pathological process of neurotoxicity. Polyphenolics have proven antioxidant activity, thereby offering protection against oxidative stress. In this study, we have isolated the polyphenolics from *Acacia nilotica* and investigated its effect against arsenic-induced neurotoxicity and oxidative stress in mice. *Acacia nilotica* polyphenolics prepared from column chromatography of the crude methanol extract using diaion resin contained a phenolic content of 452.185 ± 7.879 mg gallic acid equivalent/gm of sample and flavonoid content of 200.075 ± 0.755 mg catechin equivalent/gm of sample. The polyphenolics exhibited potent antioxidant activity with respect to free radical scavenging ability, total antioxidant activity and inhibition of lipid peroxidation. Administration of arsenic in mice showed a reduction of acetylcholinesterase activity in the brain which was counteracted by *Acacia nilotica* polyphenolics. Similarly, elevation of lipid peroxidation and depletion of glutathione in the brain of mice was effectively restored to normal level by *Acacia nilotica* polyphenolics. Gallic acid methyl ester, catechin and catechin-7-gallate were identified in the polyphenolics as the major active compounds. These results suggest that *Acacia nilotica* polyphenolics due to its strong antioxidant potential might be effective in the management of arsenic induced neurotoxicity.

## 1. Introduction

Arsenic contamination of groundwater is a global problem affecting many countries including Bangladesh, India, Pakistan, China, Argentina and USA. This problem has become much more severe in Bangladesh as 54 districts out of 64 are exposed to arsenic poisoning [1,2]. According to an estimate, approximately 35 million people have been suffering from health problems due to arsenic-contaminated drinking water, because they are exposed to an arsenic concentration in drinking water exceeding the maximum allowable limit (>50 μg/L) in drinking water in Bangladesh [3,4]. Although arsenic has been removed from the drinking water by technological means, foods chain cultivated by ground water contaminated with arsenic have remained to be another important sources of arsenic [5]. Chronic exposure of arsenic leads to arsenicosis, skin lesion, cancer, diabetes, cardiovascular disorders and neurotoxicity [6,7]. Although arsenic is associated with many different diseases and toxicities, arsenic-induced neurotoxicity has received much attention. Epidemiologic studies have shown a relationship between arsenic exposure and the neurobehavioral abnormalities in human which include learning, recent memory and attention [8,9,10]. Poorer cognitive functions including poorer memory skills, deficits in verbal skill and decrease in the IQ has been observed among children of the different parts of the world exposed to groundwater arsenic [11,12,13]. In experimental rats, Biswas et al. have shown the impairment in cognitive function following treatment with arsenic [14]. Despite the serious toxicities from arsenic exposure, there is no effective, safe and affordable treatment of this health problem.

Much effort has been directed toward elucidation of the molecular basis of neurotoxicity. Numerous studies indicated that acetylcholinesterase is a specific target of arsenic [15,16]. Acetylcholinesterase is an important component of the cholinergic system that regulates the metabolism of acetylcholine and maintain the level of acetylcholine at the cholinergic synapse. It also contributes to the synaptic plasticity. Ali et al. found an inverse relation between the level of arsenic and acetylcholinesterase among the patients of arsenic patients [15]. Mice treated with arsenic have consistently shown a decreased level of AChE associated with impairment in learning and memory [14,17]. Therefore, acetylcholinesterase is considered as a potential biomarker of arsenic induced neurotoxicity.

Oxidative stress has been recognized as the prime cause of neurotoxicity. It is evident that arsenic can generate ROS in cells such as hydroxyl radicals, superoxide anions, hydrogen peroxide and nitric oxide. As a result, cells under oxidative stress display various dysfunctions due to lesions caused by reactive oxygen species (ROS) to lipid peroxidation, oxidation of proteins, enzymes as well as DNA along with decrease in superoxide dismutase, catalase glutathione peroxidase and reduced glutathione levels [18,19,20,21]. Therefore, the recent strategy in the management of arsenic toxicity is the use of antioxidant to combat arsenic-induced oxidative stress-mediated tissue damage [22,23]. Natural antioxidants such as catechin, quercetin and curcumin have shown protective effects against arsenic induced neurotoxicity [24,25,26]. Polyphenols are naturally occurring antioxidants that confer many health benefits. Intake of phenolics has been associated with reduction ROS-induced diseases [27]. Mounting evidences suggest that polyphenols from different dietary sources are able to attenuate neurotoxicity in neurodegenerative diseases [28]. Because of the multiple pharmacological properties and the ability to cross the blood brain barrier, there is growing interest in the development of polyphenols from plants for the treatment of neurotoxicity.

*Acacia nilotica*, belonging to the family of Mimosaceae, is a tree and grows in tropical and subtropical countries, in particular is quite abundant in India, Pakistan and Bangladesh. Traditionally, the plant has been used for centuries to treat diarrhea, fever, tuberculosis and toothache [29,30]. The plant is reported to exhibit various pharmacological properties including anti-inflammatory, anti-microbial, anti-platelet and anti-carcinogenic activities [31,32,33,34]. Phytochemical investigation demonstrated that *A. nilotica* pod is a potential source of polyphenols having antioxidant activity. A large amount of polyphenols have been reported in leaves and pods [35]. Major compounds isolated from this plant are the derivatives of gallic acid and catechin [36]. These compounds have also been found in other species of Acacia and few other plants such as tea, *Loranthus globosus* and showed potential antioxidant and neuroprotective properties [37,38]. The high content of polyphenolics and antioxidant activity of *A. nilotica* led us to hypothesize that the *A. nilotica* polyphenolics might be able to counteract the oxidative stress and oxidative stress-induced damage caused by arsenic. Therefore, the present work was planned to isolate the polyphenolics with strong antioxidant activity from *A. nilotica* and evaluate its effect against arsenic induced neurotoxicity and oxidative stress in mice. Attempts were also taken to identify the compounds responsible for activity.

## 2. Materials and Methods

### 2.1. Chemicals

Sodium arsenite, diaion resin, acetylthiocholine iodide, DPPH (2,2′-diphenyl-1-picrylhydrazyl), aluminum chloride, Folin–Ciocalteu reagent, ammonium molybdate, thiobarbituric acid (TBA), tricholoroacetic acid (TCA), 2-deoxy-D-ribose, 5,5′-dithio-bis-(2-nitro) benzoic acid (DTNB), potassium ferricyanide and Tris-HCl were procured from Sigma-Aldrich, Taufkirchen, Germany. Catechin, gallic acid and ascorbic acid were obtained from Wako Pure Chemical Company Ltd., Osaka, Japan. Methanol, toluene and ethylacetate were purchased from Active Fine Chemicals Limited, Dhaka, Bangladesh. All other chemicals, unless specified, were of analytical grade.

### 2.2. Plant Sample

The ripe fruits of *Acacia nilotica* were amassed from the campus of Rajshahi University, Bangladesh in the month of June, 2017 and authenticated by an expert taxonomist. The voucher specimen (collection number 102) was deposited at the herbarium of Botany Department of Rajshahi University.

### 2.3. Extraction

The pods, after washing with distilled water, were dried for several days in an oven at 45 °C and then ground into a coarse powder by a grinding machine. The powder (500 gm) was soaked in methanol (1.4 L) an amber colored bottle and stirred occasionally for about 4 days. The resulting mixture was filtered through Whatman no.1 filters paper and was concentrated under reduced pressure at 50 °C temperature by a rotary evaporator to obtain the crude methanolic extract (CME, 29 gm).

### 2.4. Preparation of Polyphenolics (ANPP)

The CME, so obtained, was subjected to diaion resin column to separate the polyphenolics. At first, sufficient amount of distilled water was added to the CME, stirred gently for several minutes and filtered to remove the water insoluble and fatty materials. Then the aqueous part was passed through the diaion resin column, and eluted with methanol to separate the polyphenolics (ANPP). Finally, the methanol was evaporated, and the extract was freeze dried to obtain powdered polyphenolics.

### 2.5. Quantitation of Phenolics and Flavonoids of ANPP

#### 2.5.1. Total Phenolic Content (TPC)

The TPC of ANPP was estimated by the Folin–Ciocalteu (FC) method with gallic acid as standard [39]. First, 1 mL of ANPP at different concentration was added to 5 mL of FC solution in a test tube. Afterward, 5 mL solution of sodium carbonate (7.5%) was mixed and kept at room temperature for 20 min. The mixture was analyzed by a spectrophotometer at 760 nm against a blank. The result was calculated from the standard curve of gallic acid and expressed as mg equivalent of gallic acid/gm dried sample.

#### 2.5.2. Total Flavonoid Content (TFC)

The TFC of ANPP was estimated by the aluminum chloride method with catechin as standard as described earlier [40]. Catechin was used as standard, and the flavonoid content of the extracts were expressed as mg of catechin equivalent/gm of dried extract. First, 1 mL of ANPP was taken in a test tube to which 3 mL of methanol, 0.2 mL aluminum chloride (10%), 0.2 mL potassium acetate (1 M) and 5.6 mL distilled water were added. After incubation for half an hour, the mixture was analyzed by a spectrophotometer at 760 nm against a blank. The result was calculated from the standard curve of catechin and expressed as mg equivalent of catechin/gm dried sample.

### 2.6. Antioxidant Activity of ANPP

The antioxidant activity of ANPP was assessed by DPPH and hydroxyl radical scavenging, total antioxidant activity and lipid peroxidation inhibition assays.

#### 2.6.1. DPPH Radical Scavenging Activity

The DPPH radical scavenging ability was determined following the method as described earlier [41]. Briefly, 0.135 mM DPPH was prepared in methanol. In 3 mL of DPPH solution, 2 mL of ANPP at several concentrations was added and the mixture was kept at room temperature for half an hour. The solution was analyzed by spectrophotometer at 517 nm and the percent scavenging activity was computed using the following formula:
[(A _absorbance of control_ − A _absorbance of sample_) / A _absorbance of control_] × 100(1)

#### 2.6.2. Hydroxyl Radical Scavenging Activity

The scavenging activity of ANPP against hydroxyl radical was evaluated by the deoxyderibose degradation method [42]. Briefly, a reaction mixture was prepared using 2.8 mM 2-deoxy-D-ribose. 20 mM potassium dihydrogen phosphate buffer (pH 7.4), 100 µM ferric chloride, 100 µM ethylene diamine tetraacetic acid, 1 mM hydrogen peroxide and 100 µM ascorbic acid. To 1 mL of reaction mixture, ANPP at several different concentration was added and incubated at 37 °C for an hour. Then, 0.5 mL of the resulting solution was mixed with 1 mL of TCA (2.8%) and TBA (1%). The solution was analyzed by spectrophotometer at 532 nm and the percent scavenging activity was computed using the following formula:
[(A _absorbance of control_ − A _absorbance of sample_) / A _absorbance of control_] × 100(2)

#### 2.6.3. Total Antioxidant Capacity (TAC)

The TAC of ANPP was determined following the method as described [43] with catechin as standard. Briefly, ANPP at a concentration of 10–100 µg/mL was mixed with 28 mM sodium phosphate, 0.6 M sulfuric acid and 4 mM ammonium molybdate and heated for 90 min at 95 °C. The solution was analyzed by spectrophotometer at 695 nm.

#### 2.6.4. Determination of Lipid Peroxidation Inhibition Activity

ANPP was assayed for inhibition against peroxidation of lipid following the method as described [44]. Brain lipid was prepared from mouse brain through homogenization of the brain in 50 mM phosphate buffer (pH 7.4) containing 0.15 M KCl and centrifugation at 10,000× *g* for 15 min at 4 °C. ANPP at a concentration of 3.125–50 µg/mL was mixed with 0.5 mL brain lipid and 0.2 mM ferric chloride and incubated at 37 °C for half an hour. Following addition of 2 mL HCl (0.25 N) containing TCA (15%), TBA (0.38%) and BHT (0.5%), the mixture was heated at 80 °C for an hour. The solution was analyzed by spectrophotometer at 532 nm and the percent inhibition of lipid peroxidation was computed using the following formula:
[(A _absorbance of control_ − A _absorbance of sample_) / A _absorbance of control_] × 100(3)

### 2.7. In Vivo Study

The in vivo study was carried out to evaluate the biochemical effects of ANPP against arsenic induced toxicity in mice following method as described earlier [45].

#### 2.7.1. Experimental Animals and Treatment

Adult Swiss Albino mice (male) were collected from the International Centre for Diarrhoeal Disease Research, Bangladesh (ICDDR, B) and housed in congenial conditions for seven days for adjustment and suitable food and water were provided ad libitum according to the formula given by ICDDR, B. The experimental protocol was approved (Ethical clearance number: 102) by the Institutional Animal, Medical Ethics, Biosafety and Biosecurity Committee (IAMBBC) at the Institute of Biological Science, University of Rajshahi. Mice was assigned into four equal groups, each group consists of six mice. Group 1, as control group, received normal saline for 21 days. Group 2 received ANPP 50 mg/kg/day p.o. for 14 consecutive days. Group 3 received sodium arsenite 10 mg/kg/day p.o. for 21 days. Group 4 and 5 received sodium arsenite for 7 days prior to ANPP (25 and 50 mg/kg/day, respectively) treatment and continued up to 21 days simultaneously with sodium arsenite administration (14 days). All administrations were performed orally by gavage.

#### 2.7.2. Biochemical Assays

Brain sample collection. After the administration of the last dose, mice were sacrificed by cervical dislocation after anesthetized with sodium pentobarbital (30 mg/kg; intraperitoneal injection) and brain was collected. It was then washed with saline and homogenized with phosphate buffer (0.1 mM, pH 7.4). The resulting mice brain homogenate (MBH) was centrifuged at 5000 rpm for 20 min to obtain the supernatant and used for biochemical test.

Estimation of protein concentration: The protein concentration of brain homogenate was determined by the Lowry et al. (1951) procedure employing bovine serum albumin calibration curve [46]. The results were presented as mg of protein present per mL of mice brain homogenate (MBH).

Determination of brain acetylcholinesterase (AChE) activity: The AChE enzyme activity in mice brain homogenate was determined by Ellman et al. method (1961) using acetylthiocholine iodide as substrate [47]. The rates of hydrolysis by acetylcholinesterase were monitored spectrophotometrically. First, 200 µL MBH solution was incubated with 3.5 mL of reaction mixture containing acetylthiocholine iodide (0.5 mM), 5, 5′-dithio-bis (2-nitro benzoic acid) (1 mM) in phosphate buffer (50 mM, pH 8.0) at 37 °C for 15 min. The rates of hydrolysis by AChE was measured by spectrophotometer at 405 nm for 10 min at 2 min intervals. The activity of AChE was expressed as specific activity (mU/mg).

Measurement of brain MDA level. The level of MDA was measured by TBA assay method [48]. First, 0.1 mL MBH was added to 1.5 mL acetic acid (20%; pH 3.5), 1.5 mL TBA (0.8%) and 0.2 mL SDS (8.1%), and heated at 100 °C for an hour. The mixture was vigorously mixed with 5 mL *n*butanol:pyridine (15:1) and 1 mL distilled water and centrifugation at 2000× *g* for 10 min to obtain the organic layer. The solution was analyzed by spectrophotometer at 532 nm. The concentration of MDA was expressed as nmol/mg protein.

Measurement of brain reduced glutathione (GSH) level. The GSH level in MBH was estimated by the Ellman (1959) method [49]. The assay was based on the reduction of 5,5-dithio-bis-2-nitrobenzoic acid by GSH resulting in yellow colored compound which was monitored at 412 nm by spectrophotometer. The level of GSH was expressed as mmol/mg protein.

### 2.8. Isolation and Characterization of Compounds from ANPP

Isolation of compounds from ANPP was performed following the method as described earlier [50]. ANPP was first chromatographed in an open column with silica gel 60 (Merck, Germany) as a stationary phase and eluted with gradient system of n-hexane, chloroform and methanol in increasing order of polarity. All fractions were checked by thin layer chromatography under UV light and the fractions with similar R_f_ values were combined. Six subfractions (A to F) were obtained of which A, C and D showed high antioxidant activity. Fraction A showed a single spot indicating representing a single compound **1** (50 mg). Fraction C and D were subjected to preparative thin layer chromatography (PTLC) on silica gel GF_254_ with ethylacetate:toluene:methanol:formic acid (2.3:3.5:1.2:0.8) as the mobile phase that afforded compound **2** (25 mg) and compound **3** (7 mg).

The compound **1, 2** and **3** were dissolved in deuterated methanol and analyzed by a Jeol-Ex 400 MHz spectrometer for ^1^H- and FT-NMR 100 MHz spectrometer for ^13^ C-NMR spectra. The structure of the compounds was identified by comparison of their spectral data with the published values [37,50].

Gallic acid methyl ester (**1**): ^1^H NMR- δ7.05 (2H, s, H-2,6), δ3.81 (3H, s); ^13^C NMR- 110.12 (C-2, C-6), δ121 (C-1), δ139.78 (C-4), δ146.43 (C-3, C-5), δ169.15 (C-1′), δ52.37 (C-2′).

Catechin (**2**): ^1^H NMR- δ: 6.86 (1H, d, J = 1.6 Hz, H-2′), 6.75 (1H, d, J = 8 Hz, H-5′), 6.73 (1H, dd, J = 2.0, 7.8 Hz, H-6′), 5.95 (1H, d, J = 2.4 Hz, H-8), 5.88 (1H, d, J = 2.0 Hz, H-6), 4.57 (1H, d, J = 7.6 Hz, H-2), 3.99 (1H, m, H-3), 2.84 (1H, dd, J = 5.6, 5.6 Hz, H-4b), 2.49 (1H, dd, J = 8.0, 8.0 Hz, H-4a); ^13^C NMR- δ: 156.4 (C-7), 156.2 (C-5), 155.5 (C-9), 144.9 (C-3′), 144.8 (C-4′), 130.9 (C-1′), 118.6 (C-6′), 114.7 (C-5′), 113.9 (C-2′), 99.5 (C-10), 94.9 (C-6), 94.1 (C-8), 81.4 (C-2), 67.4 (C-3), 27.1 (C-4).

Catechin-7-gallate (**3**): ^1^H NMR- δ7.17 (2H, s, H-2′′, H-6′′), δ6.23 (1H, d, H-8, J = 2 Hz), δ6.20 (1H, d, H-6, J = 2 Hz), δ2.93 (1H, dd, H-4, J = 5.2, 17.8 Hz), δ2.60 (1H, dd, H-4, J = 7.8, 17.8 Hz), δ4.07 (1H, H-3, m), δ4.67 (1H, d, H-2, J = 5.2 Hz), δ6.85 (1H, d, H-2′, J = 1.7 Hz), δ6.79 (1H, d, H-5′ J = 8 Hz), δ6.74 (1H, dd, H-6′, J = 8,1.7 Hz).

### 2.9. Statistical Analysis

The results were expressed as mean ± SD. Data were analyzed by Graph Pad Prism (version 8.0.1) and Microsoft Excel 2010 were used. T-test was applied to determine the significant variations (*p*-value < 0.05) between the average values. The IC_50_ values of ANPP and its compounds were computed using non-linear regression analysis in Graph Pad Prism—8.0.1.

## 3. Results

### 3.1. Isolation of Polyphenolics (ANPP)

ANPP was isolated from the *Acacia nilotica* pods powder by extraction with methanol followed by column chromatography with diaion resin. Diaion resin is commonly used to separate the polyphenolic compounds. Quantitative analysis of ANPP revealed that it contained a total phenolic content of 452.185 ± 7.879 mg GAE/gm of dried sample and total flavonoid content of 200.075 ± 0.755 mg CE/gm of dried sample (Table 1). Whereas, the phenolic content and flavonoid contents of crude methanol extract were 256.752 ± 10.086 mg GAE/gm of dried sample and 59.509 ± 1.612 mg CE/gm of dried sample, respectively. These data indicated that the chromatography with diaion resin has increased 1.7-fold phenolic content and 3.3-fold flavonoid content. 

### 3.2. Antioxidant Potential of ANPP

#### 3.2.1. DPPH Radical Scavenging Activity

The DPPH model scavenging assay was used for evaluating the antiradical activity of the ANPP. DPPH is a synthetic free radical whose color changes from its deep violet color to light yellow color upon reduction by a hydrogen donating antioxidant. In this study, ANPP resulted in discoloration of DPPH which was quantitated by the spectrophotometer. The result of DPPH scavenging activity of ANPP has been shown in Figure 1. ANPP was found to strongly scavenge the DPPH free radicals in a dose dependent manner. The IC_50_ value of the polyphenolics was 1.741 ± 0.001 μg/mL, which was 1.4 times higher than the standard antioxidant catechin (IC_50_; 2.459 ± 0.002 μg/mL). This result indicated the potential radical scavenging activity of ANPP.

#### 3.2.2. Hydroxyl Radical Scavenging Activity

Hydroxyl radical is the most toxic free radicals generated in the biological system. We evaluated the ability of ANPP to scavenge the hydroxyl radical by the deoxyribose degradation method and the result has been presented in the Figure 2. Hydroxyl radicals were generated in a Fenton non-enzymatic reaction in vitro that attacked deoxyderibose resulting in the formation of pink color, which was monitored by the spectrophotometer. ANPP was found to scavenge the hydroxyl radical with an IC_50_ value of 3.930 ± 0.146 μg/mL, which was greater than the methanol extract. Similar to DPPH scavenging, ANPP was found to exhibit potent activity than that of the reference standard catechin whose IC_50_ was found to be 7.394 ± 0.108 μg/mL under the same experimental condition. These results demonstrated that ANPP is strong scavenger of hydroxyl radical.

#### 3.2.3. Total Antioxidant Activity

The total antioxidant activity is a measure of antioxidant property. We estimated the total antioxidant activity of ANPP on the basis of its ability to reduce Mo^5+^ to Mo^6+^ and the result has been shown in the Figure 3. ANPP exhibited strong total antioxidant activity and the activity was increased with the increase of the concentration. At 100 μg/mL concentration, the absorbance of ANPP was found to be 1.831 ± 0.061.

#### 3.2.4. Lipid Peroxidation Inhibitory Activity

Arsenic induced free radicals are reported to attack unsaturated fatty acid of the cell membrane resulting in peroxidation of lipid [21]. The capacity of ANPP to inhibit the peroxidation of lipid was measured by the thiobarbituric acid (TBA) method. In this assay, lipids were prepared from mouse brain and its peroxidation was induced by ferrous ion which was monitored by the spectrophotometer. A significant amount of lipid peroxides were produced in the control sample which was dose dependently inhibited by ANPP. The result has been shown in the Figure 4. ANPP exhibited an IC_50_ value of 41.907 ± 1.052 μg/mL. Under the same experimental condition, the standard antioxidant catechin showed an IC_50_ of 12.860 ± 0.181 μg/mL. These results suggest that ANPP, due to its considerable antioxidant activity, might play a role in natural defense through prevention of lipid peroxidation caused by free radicals.

### 3.3. Effect of ANPP on Arsenic-Induced Neurotoxicity and Oxidative Stress in Mice

#### 3.3.1. ANPP Reverses the Level of AChE in Brain Induced by Arsenic in Mice

AChE is an important component of cholinergic neuronal system which is affected by arsenic toxicity. Decline of AChE is considered to be a marker of arsenic-induced neurotoxicity [17]. In this study, AChE was significantly decreased in the brain with arsenic treated animals in comparison to control animals. ANPP treatment significantly increased the reduced level of AChE (Figure 5).

#### 3.3.2. ANPP Reverses the Level of Lipid Peroxidation Induced by Arsenic in Mice

Lipid peroxidation is a common feature of patients with arsenic toxicity, and thus considered as a marker of oxidative stress [21]. Arsenic administration for 21 days caused a significant increase of lipid peroxidation (*p* < 0.05) as indicated by the increase of MDA level in the brain of mice compared with control mice (Figure 6). The increased level of peroxidation confirmed the development of oxidative stress which was similar to those reported earlier [25,26]. Co-treatment with ANPP showed a significant reduction of the elevated lipid peroxidation (*p* < 0.05).

#### 3.3.3. ANPP Reverses the Level of GSH Induced by Arsenic in Mice

GSH is reported to bind directly with arsenic causing its depletion in tissues [21]. A significant reduction of GSH level was observed in the brain (*p* < 0.05) of mice due to treatment with arsenic as compared to controls (Figure 7). Co-treatment with arsenic and ANPP increased the level of GSH in the brain (*p* < 0.05) as compared to those treated with arsenic alone suggestive a protective effect of ANPP against oxidative insult. There were no significant effect on the activity of GSH in the brain of mice treated with ANPP alone as compared to control.

### 3.4. Identification of the Constituents of ANPP and Assessment of Their Activities

To identify the compounds responsible for activity we investigated ANPP by chromatographic methods. Three major compounds were isolated from ANPP, and their structures were characterized by spectral analyses. The compounds were identified as gallic acid methyl ester (**1**) catechin, (**2**) and catechin-7-gallate (**3**) based on their ^1^H-NMR and ^13^C-NMR spectra (Figure 8, Appendix A) [36,51]. All the compounds exhibited strong antioxidant activity (Table 2).

## 4. Discussion

Neurotoxicity, among the different toxicities, is a serious health problem of the people chronically exposed to arsenic. Psycho behavioral changes, disturbances in cognition and memory, reduced IQ are the sign and symptoms of arsenic induced neurotoxicity [11,12,13]. Fetal brain is more susceptible to arsenic toxicity [52]. Oxidative stress has been found to play the crucial role in the development of neurotoxicity [18,19,20,21]. Therefore, an agent or agents which will counteract the oxidative stress, would be effective in the prevention and treatment of neurotoxicity. Polyphenols are natural antioxidants that are associated with potential antioxidant activity and found effective against oxidative stress induced diseases [53]. Plant derived polyphenols are reported to display different biological activities including hepatoprotective, carcinogenic and neuroprotective properties [54]. *Acacia nilotica* is a folk medicinal plant used in the treatment of various diseases. It is a rich source of polyphenolics. Herein, we report for the first time that ANPP, due to its strong antioxidant activity, reverses the level of biochemical parameters linked to the arsenic neurotoxicity.

Phenolics are secondary metabolites of plants that contribute to the defensive function. Polyphenolics contain hydroxyl group that have the potential to scavenge free radicals, superoxide anions, singlet oxygen and lipid peroxyradicals involved in the oxidative processes [38]. *Acacia nilotica* has been investigated earlier for the phenolic content and antioxidant activity. Sadiq et al. [35] reported the gallic acid and catechin content in the ethanol extract of the pods. Kalavani et al. [31] also found a high phenolic content from the leaves of the plant. In this study we have been able to isolate a large content of polyphenols (phenolics and flavonoids) ANPP from the plant pods through extraction with methanol followed by chromatography with diaion resin (Table 1). Interestingly, the content of flavonoids in ANPP was much higher than any other extracts reported earlier from this plant [31,34,35]. The presence of high content of polyphenolics suggests that ANPP might have potential antioxidant activity.

Previous studies have shown the antioxidant activity of the *A. nilotica* extracts [31,32,33,34,35,36,37]. In the present study, we evaluated the antioxidant potential of ANPP in several in vitro models such as DPPH and hydroxyl radical scavenging, total antioxidant activity and lipid peroxidation inhibition. DPPH radical scavenging assay is the most common method for assessment of antioxidant activity due to its simplicity and reliability [41,50]. DPPH is a stable violet colored radical that upon reduction by electron or hydrogen of the antioxidant is converted to yellow. ANPP strongly scavenged the DPPH radicals than the respective standard antioxidant catechin (Figure 1). In the biological system, hydroxyl radical is the most toxic free radical that can damage every biomolecule in living cells. We evaluated the ability of ANPP to scavenge hydroxyl radical generated in a non-enzymatic system and found that ANPP can significantly scavenge free radical in a dose dependent manner than the standard catechin (Figure 2). The radical scavenging activity of ANPP was found to be higher than the other extracts of this plant reported earlier [31,34,35]. Our result revealed that ANPP is a strong radical scavenger that may provide protection against free radicals induced by arsenic. In total antioxidant activity assay, which is based on electron donating ability of the antioxidant, ANPP significantly increased the antioxidant activity and the activity was increased with increasing concentration (Figure 3). Lipid peroxidation is the consequence of free radical attack on membranes and other lipid constituents [21]. We used brain lipid due to its high content and peroxidation was induced by ferrous sulfate. Our result (Figure 4) revealed a significant lipid peroxidation inhibitory activity of ANPP. The activity of ANPP against peroxidation of lipid resembles the activity from the ethanol extract of this plant which was prepared by sequential extraction method [31]. These results reveal that ANPP has strong antioxidant activity which might be effective in the protection against oxidative stress.

The neurotoxicity upon chronic exposure of arsenic has been studied extensively. Biswas et al. [14] have shown that the impairment in learning and memory are correlated with arsenic induced decline of acetylcholinesterase in mice. A similar relation of acetylcholinesterase with learning and memory deficit has been observed in the neurodegenerative disorder Alzheimer’s disease [55]. Therefore, acetylcholinesterase is regarded as a good indicator for the assessment of arsenic induced neurotoxicity. In this study, we found a significant reduction of acetylcholinesterase activity in arsenic treated mice, which is consistent with the previous reports (Figure 5) [17,56]. ANPP prevented the arsenic-induced reduction of acetylcholinesterase in the brain of mice. The neuroprotective effect of ANPP was very similar to that of the polyphenoloic compounds curcumin, quercetin and catechin [24,25,26]. These results suggest that ANPP has potential ability to reverse the reduction of acetylcholinesterase in arsenic treated mice.

Oxidative stress has been recognized as the important pathway of neurotoxicity. Numerous studies have shown that arsenic elevates the oxidative stress, through the generation of free radicals, and cause oxidative damage to the neuronal cell [18,19,20,21]. CNS is particularly susceptible to damage by free radical due to high content of lipids. Neuronal membrane is oxidized into peroxide. Enhanced lipid peroxidation has been observed in patients with arsenic toxicity [21]. In this study, we have found a significant elevation of lipid peroxidation in the brain of arsenic treated mice as compared to the untreated normal mice (Figure 6) which is consistent with the reports earlier [22,23]. ANPP significantly inhibited the peroxidation of lipid which is in accordance with the in vitro studies. This result could be attributed to the radical scavenging activity of ANPP.

The thiol-based non-enzyme antioxidant system play as the second line defense against oxidative damage. GSH is considered to be the biological antioxidant that are abundant in the brain tissue and play an essential role in the protection of neuron against oxidative damage. It can scavenge the free radical as well as detoxifying xenobiotics. Previous reports indicated that GSH can bind to GSH, causing its depletion in tissue [22,23]. We also observed a decrease in GSH level in mouse brain following exposure of arsenic, conforming the previous reports (Figure 7). ANPP treatment prevented the decrease of GSH and the consequent oxidative damage to neuron.

The polyphenolic compounds present in ANPP were investigated to obtain insights into their contributions to the bioactivities. Bioactivity guided separation approach led to the isolation of three major compounds from ANPP, and they were characterized by studies of their ^1^H and ^13^C spectra (Figure 8). The compounds were identified as gallic acid methyl ester and catechin-7-gallate. All the three compounds showed strong antioxidant activity (Table 2). Catechin is a flavonoid antioxidant isolated from tea and coffee as the major constituent that have neuroprotective potential in oxidative stress mediated neurodegeneration in AD mouse model [57]. Methylgallate is another phenolic compound present in food with antioxidant and neuroprotective potential [58]. Kalaniva et al. [32] isolated ethylgallate from this plant that showed multiple biological activities including antioxidant and cytotoxic activities. Catechin-7-gallate is a polyphenol present in several plants, mainly in green tea. Previous reports with catechin-7-gallate have revealed that it is a powerful antioxidant [36]. The potential activity of the identified compounds suggests that they are responsible for the activities of the ANPP.

## 5. Conclusions

In the present study, we report the isolation of polyphenolics ANPP from *A. nilotica* pods that possess strong antioxidant activity. ANPP ameliorated the biochemical markers of arsenic induced neurotoxicity and oxidative stress in mice as indicated by an increase of acetylcholinesterase activity and GSH level as well as decrease of lipid peroxidation in the arsenic exposed brain. The activities of ANPP might be due to the presence of large content of phenolic and flavonoid compounds which are attributable to antioxidant activity. Three major compounds methyl gallate, catechin and catechin-7-gallate were identified in the ANPP that exhibited the potent antioxidant activity. Hence, ANPP represents a source of potential antioxidants that may be used in the management of arsenic induced neurotoxicity and oxidative stress. However, further studies are needed to explain the mechanism behind the neuroprotective effect of ANPP.

## Figures and Tables

**Figure 1 molecules-27-01037-f001:**
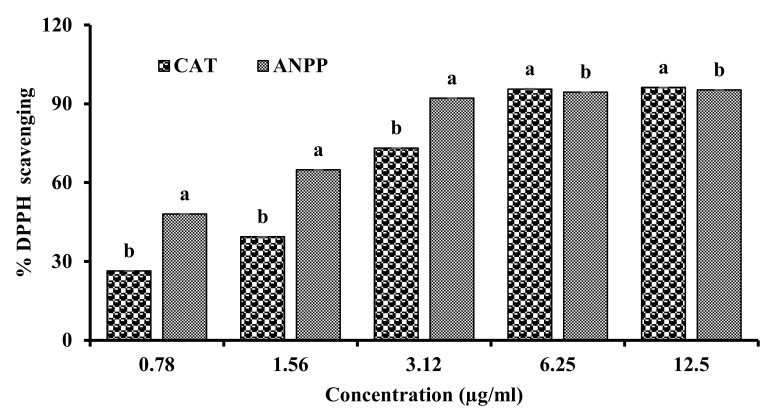
DPPH radical scavenging activity of ANPP. IC_50_ (μg/mL): CAT, 2.459 ± 0.002; ANPP, 1.741 ± 0.001. Results are expressed as mean ± SD (n = 3). Means with letters a and b differ significantly (*p* < 0.05). CAT, catechin; ANPP, *Acacia nilotica* polyphenolics.

**Figure 2 molecules-27-01037-f002:**
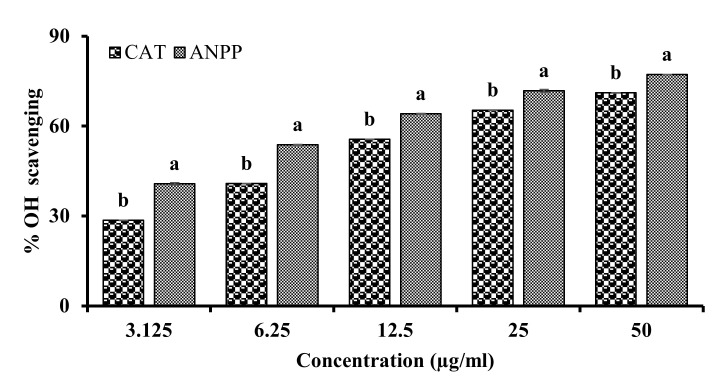
Hydroxyl radical scavenging activity of ANPP. IC_50_ (μg/mL): CAT, 7.394 ± 0.108; ANPP, 3.930 ± 0.146. Results are expressed as mean ± SD (n = 3). Means with letters a and b differ significantly (*p* < 0.05). CAT, catechin; ANPP, *Acacia nilotica* polyphenolics.

**Figure 3 molecules-27-01037-f003:**
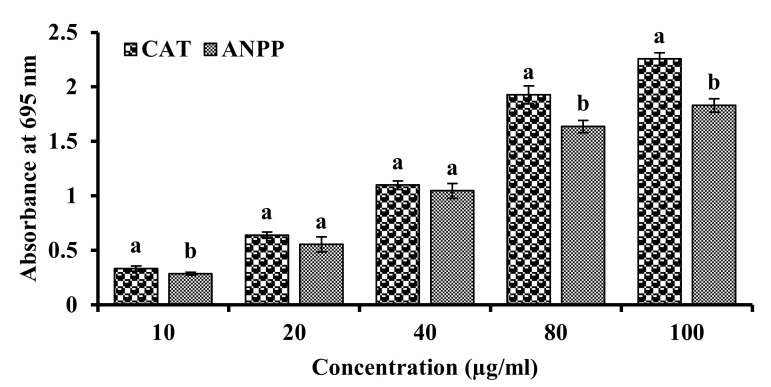
Total antioxidant activities of ANPP. At 100 μg/mL concentration, the absorbances are 2.291 ± 0.567 for CAT and 1.831 ± 0.061 for ANPP. Results are expressed as mean ± SD (n = 3). Means with letters a and b differ significantly (*p* < 0.05). CAT, catechin; ANPP, *Acacia nilotica* polyphenolics.

**Figure 4 molecules-27-01037-f004:**
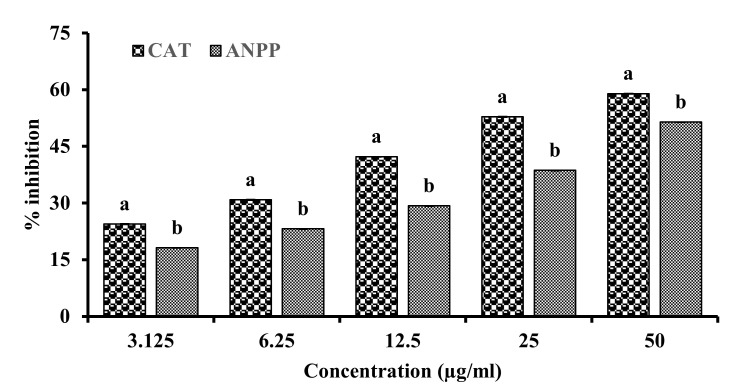
Inhibition of lipid peroxidation by ANPP. IC_50_ (μg/mL): CAT, 12.860 ± 0.181; ANPP, 41.907 ± 1.052. Results are expressed as mean ± SD (n = 3). Means with letters a and b differ significantly (*p* < 0.05). CAT, catechin; ANPP, *Acacia nilotica* polyphenolics.

**Figure 5 molecules-27-01037-f005:**
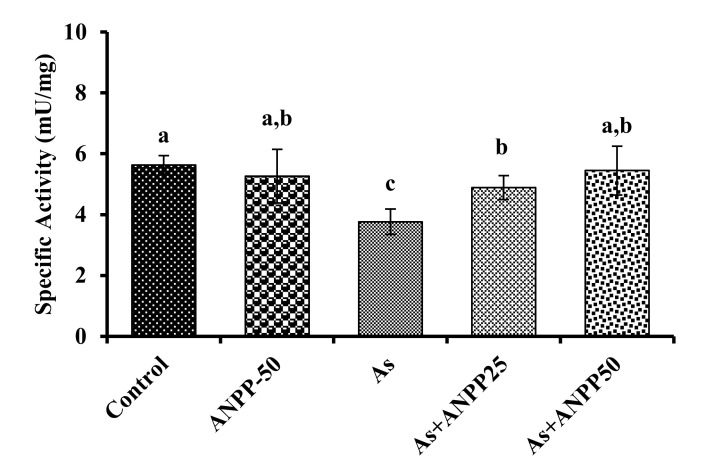
Effect of ANPP on brain AChE activity in arsenic treated mice. Means with different letters (a–c) differ significantly (*p* < 0.05), where ANPP and arsenic treated groups were significantly different (*p* < 0.05) from arsenic treated group. Values were expressed as Mean ± STD, n = 6 mice for each group. As, arsenic; ANPP25, *Acacia nilotica* polyphenolics at 25 mg/kg/day; ANPP50, *Acacia nilotica* polyphenolics at 50 mg/kg/day.

**Figure 6 molecules-27-01037-f006:**
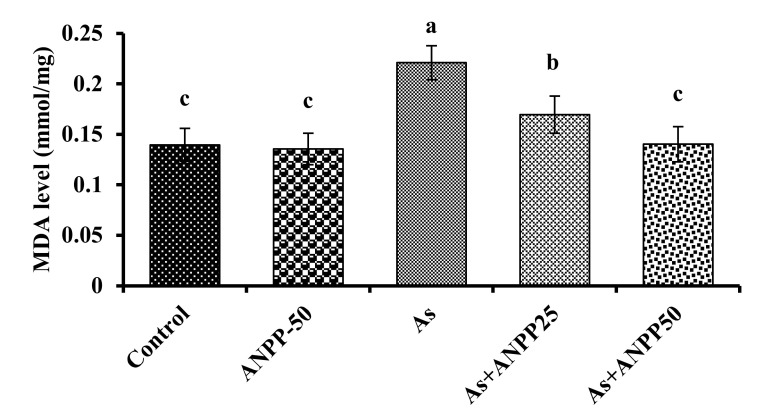
Effect of ANPP on MDA level in arsenic treated mice. Means with different letters (a–c) differ significantly (*p* < 0.05), where ANPP and arsenic treated groups were significantly different (*p* < 0.05) from arsenic treated group. Values were expressed as Mean ± STD, n = 6 mice for each group. As, arsenic; ANPP25, *Acacia nilotica* polyphenolics at 25 mg/kg/day; ANPP50, *Acacia nilotica* polyphenolics at 50 mg/kg/day.

**Figure 7 molecules-27-01037-f007:**
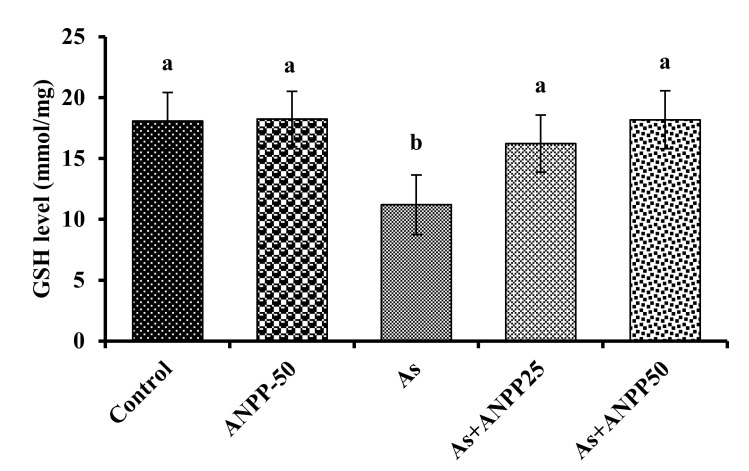
Effect of ANPP on GSH level in arsenic treated mice. Means with letters a and b differ significantly (*p* < 0.05), where all the ANPP and arsenic treated groups were significantly different (*p* < 0.05) from arsenic treated group. Values were expressed as Mean ± STD, n = 6 mice for each group. As, arsenic; ANPP25, *Acacia nilotica* polyphenolics at 25 mg/kg/day; ANPP50, *Acacia nilotica* polyphenolics at 50 mg/kg/day.

**Figure 8 molecules-27-01037-f008:**
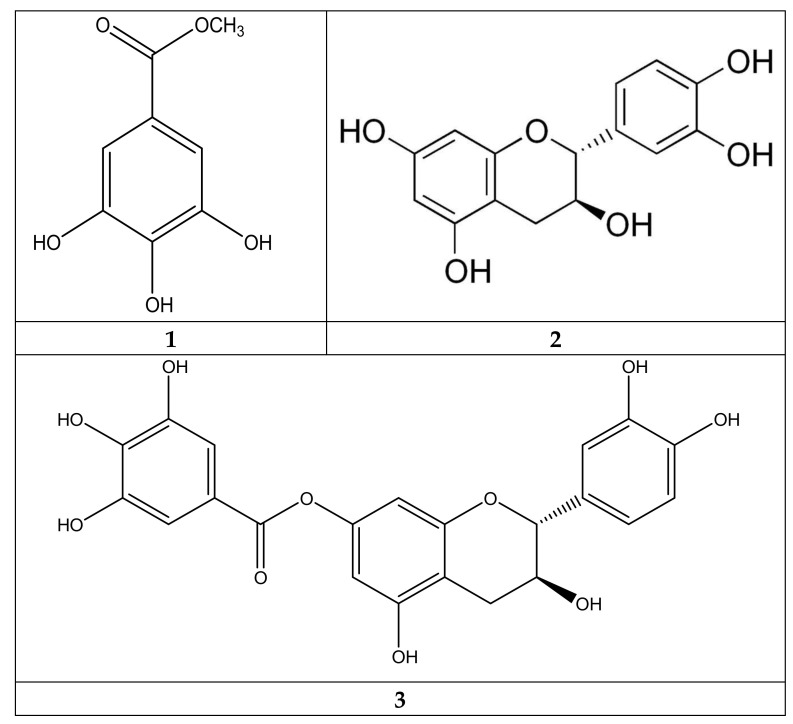
Chemical structure of the compounds **1**–**3** from ANPP.

**Table 1 molecules-27-01037-t001:** Total phenolic and flavonoid contents of the crude methanol extract and ANPP from *Acacia nilotica* pods.

Sample	TPC mg GAE/gm Dried Sample	TFC Mg CE/Gm Dried Sample
CME	256.752 ± 10.086	59.509 ± 1.612
ANPP	452.185 ± 7.879	200.075 ± 0.755

CME: crude methanolic extract; ANPP: polyphenolics of *Acacia nilotica*; TPC: total phenolic content; TFC: total flavonoid content; GAE: gallic acid equivalent; CE: catechin equivalent.

**Table 2 molecules-27-01037-t002:** IC_50_ values of the compounds **1**–**3** from for DPPH and OH radical scavenging.

IC_50_ Values (µg/mL)	Compounds		
	1	2	3
DPPH	1.047 ± 0.0546	2.558 ± 0.830	2.356 ± 0.002
OH	9.848 ± 3.674	12.100 ± 0.056	5.884 ± 0.017

## Data Availability

The data presented in this study are available on request from the corresponding author.

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
