# Peer review of "Polyphenolics with Strong Antioxidant Activity from Acacia nilotica Ameliorate Some Biochemical Signs of Arsenic-Induced Neurotoxicity and Oxidative Stress in Mice"

_molecules, 2022, doi:10.3390/molecules27031037_

Round 1
Reviewer 1 Report
I have carefully read the manuscript entitled „Polyphenolics with strong antioxidant activity from Acacia nilotica ameliorate some biochemical signs of arsenic-induced neurotoxicity and oxidative stress in mice” prepared by Foyzun et al., for publication in Molecules MDPI. The aim of this study was to report the isolation of polyphenols with strong antioxidant activity from A. nilotica pods and to test the usufulness of isolated polyphenols in the reduction of arsenic-induced neurotoxixity in mice brain tissue. In my opinion the manuscript is of relevance and will be interested for Molecules readers. However some corrections/changes will improve its quality before final publication.
Abstract – the full botanical name of the tested plant species with authors’ initials should be written. Moreover the meanings of abbreviations „GAE” and „CE” should be also explained, because they are provided in the abstract for the first time.
Keywords – Avoid using words from manuscript title as keywords (don’t repeat them, use other instead to broaden the information about your manuscript).
Introduction – lines 99-100 – the aim of the study should be more widely described and presented.
Materials and methods – information about the chemical reagents producers’ should be listed.
Results – Line 260 versus table 2 – the CME values of TCP are different between text and table.
Line 262 – folds of phenolic and flovonoid content are wrong estimated (the same for Line 276).
Line 325, 329, 337, 351, 444 – correct ANPE to ANPP.
Discussion - the entire chapter in its current form is basically a repetition of the information contained in the ‘Introduction’ and ‘Results’ chapters rather than a scientific discussion. There is no comparison and discussion of the obtained results with the results of similar studies, no references to the literature. This chapter is the weakest part of the manuscript, requiring a major rewritting.
Conclusions – the future directions of studies should be given.
Author Response
Comment 1: Abstract – the full botanical name of the tested plant species with authors’ initials should be written. Moreover the meanings of abbreviations „GAE” and „CE” should be also explained, because they are provided in the abstract for the first time.
Response 1: We have revised the manuscript accordingly.
Comment 2: Keywords – Avoid using words from manuscript title as keywords (don’t repeat them, use other instead to broaden the information about your manuscript).
Response 2: Thanks for nice suggestion. Accordingly, we have added several new keywords
Comment 3: Introduction – lines 99-100 – the aim of the study should be more widely described and presented.
Response 3: According to comment, the aim of the study has been rewritten in the lines 95-102.
Comment 4: Materials and methods – information about the chemical reagents producers’ should be listed.
Response 4: We have added the chemical information in the lines 104-112.
Comment 5: Results – Line 260 versus table 2 – the CME values of TCP are different between text and table. Line 262 – folds of phenolic and flovonoid content are wrong estimated (the same for Line 276). Line 325, 329, 337, 351, 444 – correct ANPE to ANPP.
Response 5: We have corrected accordingly.
Comment 6: Discussion - the entire chapter in its current form is basically a repetition of the information contained in the ‘Introduction’ and ‘Results’ chapters rather than a scientific discussion. There is no comparison and discussion of the obtained results with the results of similar studies, no references to the literature. This chapter is the weakest part of the manuscript, requiring a major rewritting.
Response 6: According to comment, we have revised the entire discussion section and hope the present form is much more improved.
Comment 7: Conclusions – the future directions of studies should be given.
Response 7: Future direction has been added in the revised manuscript in the lines 498-499.
Reviewer 2 Report
1. The spectral data must be checked carefully:
Gallic acid methyl ester (1): 1H NMR- δ7.05 (2H, s, H-5,6), it should not be H-5,6 rather H-2,6
2. Mechanism as antioxidant should be elaborated.
3. Introduction should be revised to highlight the importance of phenolics constituents in other related species.
4. The original spectra must be provided for better understanding.
Author Response
Comment 1: The spectral data must be checked carefully: Gallic acid methyl ester (1): 1H NMR- δ7.05 (2H, s, H-5,6), it should not be H-5,6 rather H-2,6.
Response 1: We have corrected the typo mistake. Thanks for your suggestion and efforts.
Comment 2: Mechanism as antioxidant should be elaborated.
Response 2: In the lines 408-410, we have elaborated the mechanism as antioxidant.
Comment 3: Introduction should be revised to highlight the importance of phenolics constituents in other related species.
Response 3: Thanks for suggestion. We have revised accordingly and added lines 94-96.
Comment 4: The original spectra must be provided for better understanding.
Response 4: According to your suggestion, we have provided the original spectra as supplementary files.
Reviewer 3 Report
Dear Editor
I suggest that the article cannot be accepted for publication at its present level. For publication, major revision is needed. The comments are listed as follows.
1 Language of this manuscript has to be improved.
2 The introduction part is not clear and effective. It needs to be more concise for acceptance. What is the purpose of this study?
3 To further verify the viability of the method, recovery experiments should be carried out.
4 The conclusions need to be rewritten.
5 And some of the references are too old, recently published papers should be cited. for exemple
Phytochemical screening and the antioxidant, antibacterial and antifungal activities of aqueous extracts from the leaves of Salvia officinalis planted in Morocco
I Maliki, A EL Moussaoui, M Ramdani, K ELBadaoui
DOI: https://doi.org/10.48317/IMIST.PRSM/morjchem-v9i2.24840
Antibacterial, antifungal and antioxidant activity of total polyphenols of Withania frutescens.L
Bioorganic Chemistry, Volume 93, December 2019, Article 103337
Abdelfattah El Moussaoui, Fatima Zahra Jawhari, Ahmed M. Almehdi,Amina Bari
Author Response
Comment 1: Language of this manuscript has to be improved.
Response 1: We have thoroughly checked and improved the manuscript. Hope this revised version is much more improved.
Comment 2: The introduction part is not clear and effective. It needs to be more concise for acceptance. What is the purpose of this study?
Response 2: We have revised the introduction section (lines 58-59, 76-85) and the purpose of the study has been clearly stated in the lines 95-102.
Comment 3: To further verify the viability of the method, recovery experiments should be carried out.
Response 3: We have followed the standard methods and used negative control and positive control in all the experiments to exclude the possibility of error.
Comment 4: The conclusions need to be rewritten.
Response 4: We have revised the conclusion accordingly.
Comment 5: And some of the references are too old, recently published papers should be cited. for exemple ; Phytochemical screening and the antioxidant, antibacterial and antifungal activities of aqueous extracts from the leaves of Salvia officinalis planted in Morocco
I Maliki, A EL Moussaoui, M Ramdani, K ELBadaoui
DOI: https://doi.org/10.48317/IMIST.PRSM/morjchem-v9i2.24840; Antibacterial, antifungal and antioxidant activity of total polyphenols of Withania frutescens.L
Bioorganic Chemistry, Volume 93, December 2019, Article 103337
Abdelfattah El Moussaoui, Fatima Zahra Jawhari, Ahmed M. Almehdi,Amina Bari
Response 5: We have replaced the old references where it is required (number 2 and 18).
Reviewer 4 Report
The manuscript by Foyzun et al “Polyphenolics with Strong Antioxidant Activity from Acacia nilotica Ameliorate Some Biochemical Signs of Arsenic-induced Neurotoxicity and Oxidative Stress in Mice” provides information on the antioxidant activity of Acacia nilotica. However, the authors could improve the manuscript by clarifying the following points:
Cytotoxicity studies of the ANPP extract must be carried out, to rule out that the observed effect is not due to the toxicity of the ANPP extract.
Although a fractionation of the methanol extract of A. nilotica was carried out to obtain three compounds (Gallic acid methyl ester, Catechin and Catechin-7-gallate). The authors should take into account that previous works have reported the presence of tannins (polygalloyltannin, ethyl gallate, 1-O-galloyl-β-D-glucose, 1,6-di-O-galloyl-β-D-glucose, digallic acid, gallocatechin-5-O-gallate, ellagic acid, (-) - epigallocatechin-7-gallate, (-) - epigallocatechin-5,7-digallate, dicatechin) and flavonoids (catechin-7-O-gallate, quercetin, isoquercetin, naringenin, naringenin-7-O-β-glucopyranoside, chalconaringenin-4′-O-β-glucopyranoside, kaempferol, (þ) -catechin-5-gallate, (þ) -catechin-5,7-digallate, (þ) -catechin-3′, 5-digallate, (þ) -catechin-4 ′, 5-digallate, quercitin- 3-galactosyl, (þ) -mollisacacidin, rutin, vicenin, epicatechin, melacacidin, naringenin-7- O-β-D- (6′-O-galloyl) glucopyranoside, leucocyanadin, kaempferol-7-glucoside, acacetin) in this plant species. It is for this reason that it is important to attach an analysis of the methanol extract (HPLC or NMR) to indicate the presence or absence of these compounds.
The pharmacological activities reported by the in vivo tests carried out on the ANPP extract, may be due to the compounds mentioned above and not only due to the three isolated compounds. Why weren't in vivo experiments performed on the isolated compounds, in order to compare them with the ANPP extract?
On what did the researchers base themselves to choose the doses of 25 and 50 mg / kg of weight? Why were no other doses taken? At these doses, the ANPP extract does not show toxicity?
Author Response
Comments 1: Cytotoxicity studies of the ANPP extract must be carried out, to rule out that the observed effect is not due to the toxicity of the ANPP extract.
Response 1: To rule out the toxicity of the ANPP, we used a group of mice that received the high dose 50 mg/kg body weight according to the protocol. As has been found in the Figure 5-7 there were no changes in the biochemical marker of neurotoxicity and oxidative stress. At the same time, there were no visible sign of toxicities in mice.
Comment 2: Although a fractionation of the methanol extract of A. nilotica was carried out to obtain three compounds (Gallic acid methyl ester, Catechin and Catechin-7-gallate). The authors should take into account that previous works have reported the presence of tannins (polygalloyltannin, ethyl gallate, 1-O-galloyl-β-D-glucose, 1,6-di-O-galloyl-β-D-glucose, digallic acid, gallocatechin-5-O-gallate, ellagic acid, (-) - epigallocatechin-7-gallate, (-) - epigallocatechin-5,7-digallate, dicatechin) and flavonoids (catechin-7-O-gallate, quercetin, isoquercetin, naringenin, naringenin-7-O-β-glucopyranoside, chalconaringenin-4′-O-β-glucopyranoside, kaempferol, (þ) -catechin-5-gallate, (þ) -catechin-5,7-digallate, (þ) -catechin-3′, 5-digallate, (þ) -catechin-4 ′, 5-digallate, quercitin- 3-galactosyl, (þ) -mollisacacidin, rutin, vicenin, epicatechin, melacacidin, naringenin-7- O-β-D- (6′-O-galloyl) glucopyranoside, leucocyanadin, kaempferol-7-glucoside, acacetin) in this plant species. It is for this reason that it is important to attach an analysis of the methanol extract (HPLC or NMR) to indicate the presence or absence of these compounds.
Response 2: Very good suggestion. As we found the ameliorating effect of ANPP in arsenic induced neurotoxicity in mice, we were very much interested to identify the compounds that are contributing to the activity. We were able to characterize the three major compounds and determine their roles in the activity. We are pleased to let you know that in continuation of the study, we are now in the process of characterizing the remaining compounds present in the ANPP which will be published in the next.
Comment 3: The pharmacological activities reported by the in vivo tests carried out on the ANPP extract, may be due to the compounds mentioned above and not only due to the three isolated compounds. Why weren't in vivo experiments performed on the isolated compounds, in order to compare them with the ANPP extract?
Response 3: In this study we have shown the role of three compounds in the activity of ANPP by in vitro studies. However, we have plan to characterize the remaining compounds and to study their effects in arsenic induced neurotoxicity in mice.
Comment 4: On what did the researchers base themselves to choose the doses of 25 and 50 mg / kg of weight? Why were no other doses taken? At these doses, the ANPP extract does not show toxicity?
Response 4: Based on the IC50 values of ANPP in radical scavenging and inhibition of lipid peroxidation and a review of the literature of the toxicities as well as doses of the extract of Acacia nilotica that has been tested against streptozotocin induced diabetes in mice we choose 25 and 50 mg / kg of weight for this study.
Reviewer 5 Report
I have carefully read the manuscript. It is well organized, while study was carefully conducted. I am interested in analysis of the prepared extract. Why did you use combination of TLC and NMR? HPLC would be better choice in this case.
Also, why did you use maceration as extraction technique, when it is the least efficient of all techniques?
Author Response
Comment 1: I have carefully read the manuscript. It is well organized, while study was carefully conducted. I am interested in analysis of the prepared extract. Why did you use combination of TLC and NMR? HPLC would be better choice in this case.
Response 1: The important finding of this study was the Isolation of high content of polyphenolics from Acacia nilotica with strong antioxidant activity that exhibited ameliorating effect in arsenic induced neurotoxicity in mice. To gain insights into the compounds in the ANPP that are contributing to the activity we attempted to isolate and identify the individual compounds and determined the bioactivity. For this reason we used a combination of TLC and NMR rather than HPLC.
Comment 2: Also, why did you use maceration as extraction technique, when it is the least efficient of all techniques?
Response 2: We preferred cold extraction method to avoid any breakdown of polyphenolic compounds although it is the least efficient technique.
Round 2
Reviewer 1 Report
One comment to revised version of the manuscript:
Abstract Line 25 - provide full botanical name with authors initials, Acacia niotica (L.) Delile, only in this line 25. Then, it is not necessary to use the full species name, you can use only A. nilotica.
Reviewer 4 Report
The authors of the manuscript "Polyphenolics with Strong Antioxidant Activity from Acacia nilotica Ameliorate Some Biochemical Signs of Arsenic-induced Neurotoxicity and Oxidative Stress in Mice" have addressed all comments suggested by the reviewers. In its current state, it can be published in the journal.
Reviewer 5 Report
Authors answered all issues and did the necessary changes into the manuscript. Therefore, I suggest acceptance for publication in the present form.